# Efficient Empowerment Estimation for Unsupervised Stabilization

**Ruihan Zhao, Kevin Lu, Pieter Abbeel, Stas Tiomkin**
Berkeley Artificial Intelligence Research Lab
Electrical Engineering and Computer Sciences
University of California, Berkeley, CA, USA
`{philipzhao,kzl,pabbeel,stas}@berkeley.edu`

## Abstract

Intrinsically motivated artificial agents learn advantageous behavior without externally-provided rewards. Previously, it was shown that maximizing mutual information between agent actuators and future states, known as the empowerment principle, enables unsupervised stabilization of dynamical systems at upright positions, which is a prototypical intrinsically motivated behavior for upright standing and walking. This follows from the coincidence between the objective of stabilization and the objective of empowerment. Unfortunately, sample-based estimation of this kind of mutual information is challenging. Recently, various variational lower bounds (VLBs) on empowerment have been proposed as solutions; however, they are often biased, unstable in training, and have high sample complexity. In this work, we propose an alternative solution based on a trainable representation of a dynamical system as a Gaussian channel, which allows us to efficiently calculate an unbiased estimator of empowerment by convex optimization. We demonstrate our solution for sample-based unsupervised stabilization on different dynamical control systems and show the advantages of our method by comparing it to the existing VLB approaches. Specifically, we show that our method has a lower sample complexity, is more stable in training, possesses the essential properties of the empowerment function, and allows estimation of empowerment from images. Consequently, our method opens a path to wider and easier adoption of empowerment for various applications. [1]

## 1 Introduction

Intrinsic motivation allows artificial and biological agents to acquire useful behaviours without external knowledge (Barto et al. (2004); Chentanez et al. (2005); Schmidhuber (2010); Barto (2013); Oudeyer et al. (2016)). In the framework of reinforcement learning (RL), this external knowledge is usually provided by an expert through a task-specific reward, which is optimized by an artificial agent towards a desired behavior (Mnih et al. (2013); Schulman et al. (2017)).

In contrast, an intrinsic reward can arise solely from the interaction between the agent and environment, which eliminates the need for domain knowledge and reward engineering in some settings (Mohamed & Rezende (2015); Houthooft et al. (2017); Pathak et al. (2017)). Previously, it was shown that maximizing mutual information (Cover & Thomas (2012)) between an agent's actuators and sensors can guide the agent towards states in the environment with higher potential to achieve a larger number of eventual goals (Klyubin et al. (2005); Wissner-Gross & Freer (2013)).

Maximizing this kind of mutual information is known as the empowerment principle (Klyubin et al. (2005); Salge et al. (2014)). Previously, it was found that an agent maximizing its empowerment converges to an unstable equilibrium of the environment in various dynamical control systems (Jung et al. (2011); Salge et al. (2013); Karl et al. (2019)). In this application, ascending the gradient of the empowerment function coincides with the objective of optimal control for stabilization at an unstable equilibrium (Strogatz (2018)), which is an important task for both engineering (Todorov (2006)) and

---

[1]Project page: `https://sites.google.com/view/latent-gce`

biological[2] systems; we refer to this as the *essential property* of empowerment. It follows from the aforementioned prior works that a plausible estimate of the empowerment function should possess this essential property.

Empowerment has been found to be useful for a broad spectrum of applications, including: unsupervised skill discovery (Sharma et al. (2020); Eysenbach et al. (2019); Gregor et al. (2017); Karl et al. (2019); Campos et al. (2020); human-agent coordination (Salge & Polani (2017); Guckelsberger et al. (2016)); assistance (Du et al. (2021)); and stabilization (Tiomkin et al. (2017)). Past work has utilized variational lower bounds (VLBs) (Poole et al. (2019); Alemi et al. (2017); Tschannen et al. (2020); Mohamed & Rezende (2015)) to achieve an estimate of the empowerment. However, VLB approaches to empowerment in dynamical control systems (Sharma et al. (2020); Achiam et al. (2018)) have high sample complexity, are often unstable in training, and may be biased. Moreover, it was not previously studied if empowerment estimators learned via VLBs possess the essential properties of empowerment.

In this work, we introduce a new method, Latent Gaussian Channel Empowerment (Latent-GCE), for empowerment estimation and utilize the above-mentioned property as an "indicator" for the quality of the estimation. Specifically, we propose a particular representation for dynamical control systems using deep neural networks which is learned from state-action trajectories. This representation admits an efficient estimation of empowerment by convex optimization (Cover & Thomas (2012)), both from raw state and from images. We propose an algorithm for simultaneous estimation and maximization of empowerment using standard RL algorithms such as Proximal Policy Optimization (Schulman et al. (2017)), and Soft Actor-Critic (Haarnoja et al. (2018)). We test our method on the task of unsupervised stabilization of dynamical systems with solely intrinsic reward, showing our estimator exhibits essential properties of the empowerment function.

We demonstrate the advantages of our method through comparisons to the existing state-of-the-art empowerment estimators in different dynamical systems from the OpenAI Gym simulator (Brockman et al. (2016)). We find that our method (i) has a lower sample complexity, (ii) is more stable in training, (iii) possesses the essential properties of the empowerment function, and (iv) allows us to accurately estimate empowerment from images. We hope such a review of the existing methods for empowerment estimation will help push this research direction.

## 2 PRELIMINARIES

In this section, we review the necessary background for our method, consisting of the reinforcement learning setting, various empowerment estimators, and the Gaussian channel capacity. We also review the underlying components in relevant prior work which we use for comparison to our method.

### 2.1 REINFORCEMENT LEARNING

The reinforcement learning (RL) setting is modeled as an infinite-horizon Markov Decision Process (MDP) defined by: the state space $\mathcal{S}$, the action space $\mathcal{A}$, the transition probabilities $p(s'|s, a)$, the initial state distribution $p_0(s)$, the reward function $r(s, a) \in \mathbb{R}$, and the discount factor $\gamma$. The goal of RL is to find an optimal control policy $\pi(a|s)$ that maximizes the expected return, i.e. $\max_\pi \mathbb{E}_{s_0 \sim p_0, a_t \sim \pi, s_{t+1} \sim p} \left[ \sum_{t=0}^\infty \gamma^t r(s_t, a_t) \right]$ (Sutton & Barto (2018)).

### 2.2 EMPOWERMENT

Interaction between an agent and its environment is plausibly described by the perception-action cycle (PAC), (Tishby & Polani (2011)), where the agent observes the state of the environment via its sensors and responds with its actuators. The maximal information rate from actuators to sensors is an inherent property of PAC which characterizes the *empowerment* of the agent, formally defined below.

Empowerment (Klyubin et al. (2005)) is defined by the maximal mutual information rate (Cover & Thomas (2012)) between the agent's sensor observations $o \in \mathcal{O}$ and actuators $a \in \mathcal{A}$ given the current state $s \in \mathcal{S}$. It is fully specified by a fixed probability distribution of sensor observations conditioned

---

[2]Broadly speaking, an increase in the rate of information flow in the perception-action loop (Tishby & Polani (2011)) could be an impetus for the development of *homosapiens*, as hypothesized in (Yuval (2014)).

on the actuators and the state, $p(O \mid A, s)$, denoted by "channel", and by a free probability of the actuator signals conditioned on the state, $\omega(A \mid s)$, denoted by "source". The empowerment $\mathcal{E}$ is the capacity of the channel, which is found via optimization of the source distribution $\omega(A \mid s)$:

$$\mathcal{E}(s) = \max_{\omega(A|s)} \mathcal{I}[O; A \mid s] = \max_{\omega(A|s)} \sum_{O,A} p(O, A \mid s) \log\left(\frac{p(A \mid O, s)}{\omega(A \mid s)}\right) \tag{1}$$

where $\mathcal{I}[O; A \mid s]$ is the mutual information, reflecting the difference between the entropy of the sensor inputs, $\mathcal{H}(O \mid s)$ and the corresponding conditional entropy, $\mathcal{H}(O \mid A, s)^3$.

In this work, we follow the original empowerment formulation (Klyubin et al. (2005)), where $A$ is an action sequence of length $T$ starting from the state $s$ and $O$ is the resulting observation sequence. However, our method is applicable to other choices of observation and control sequences.

**Remark 1** *The source distribution in Equation 1, $\omega^*(A \mid s)$, is used only for the estimation of empowerment. Another policy, $\pi(a \mid s)$, is calculated using empowerment as an intrinsic reward by either reinforcement learning (Karl et al. (2019)) or a 1-step greedy algorithm (Jung et al. (2011); Salge et al. (2013)) to reach the states with maximal empowerment. Strictly speaking, the source distribution $\omega$ maximizes the mutual information, (i.e. estimates the empowerment), while the policy distribution $\pi$ maximizes expected accumulated empowerment along trajectories.*

In general, the capacity of an arbitrary channel for a continuous random variable is unknown except for a few special cases such as the Gaussian linear channel, as reviewed below. As a result, past works have largely relied on variational lower bounds instead.

### 2.2.1 EMPOWERMENT BY VARIATIONAL LOWER BOUNDS

The mutual information in Equation 1 can be bounded from below as follows (Mohamed & Rezende (2015); Gregor et al. (2017); Karl et al. (2019); Poole et al. (2019)):

$$\sum_{O,A} p(O, A \mid s) \log\left(\frac{p(A \mid O, s)}{\omega(A \mid s)}\right) = \sum_{O,A} p(O, A \mid s) \log\left(\frac{q(A \mid O, s)}{\omega(A \mid s)}\right) + D_{KL}\left[p(A \mid O, s) \mid\mid q(A \mid O, s)\right]$$

$$\leq \sum_{O,A} p(O, A \mid s) \log\left(\frac{q(A \mid O, s)}{\omega(A \mid s)}\right) \tag{2}$$

where $q(A \mid O, s)$ and $\omega(A \mid s)$ are represented by neural networks with parameters $\phi$ and $\psi$ respectively. The lower bound in Equation 2 can be estimated from samples using reinforcement learning with intrinsic reward $r(s_t, a_t) = \sum_t^T \log q_\phi(a_t \mid o_t, s) - \log \omega_\psi(a_t \mid s_t)$, as detailed in (Gregor et al. (2017); Karl et al. (2019)). To apply this variational lower bound on empowerment to unsupervised stabilization of dynamical systems one needs to learn three distributions: $q_\phi(a_t \mid o_t, s)$ and $\omega_\psi(a_t \mid s_t)$ for the lower bound in Equation 2, and $\pi(a \mid s)$ as the control policy (see Remark 1).

Developing the mutual information in Equation 1 in the opposite direction, one gets another lower bound on empowerment, an approximation of which was used in (Sharma et al. (2020)):

$$\sum_{O,A} p(O, A \mid s) \log\left(\frac{p(O \mid A, s)}{\omega(O \mid s)}\right) = \sum_{O,A} p(O, A \mid s) \log\left(\frac{q(O \mid A, s)}{\omega(O \mid s)}\right) + D_{KL}\left[p(O \mid A, s) \mid\mid q(O \mid A, s)\right]$$

$$\leq \sum_{O,A} p(O, A \mid s) \log\left(\frac{q(O \mid A, s)}{\omega(O \mid s)}\right). \tag{3}$$

For sake of the clarity of the exposition, the above-explained variational empowerment is schematically depicted in Figure 1. The controller policy, $\pi_\theta(a \mid s)$, collects transitions in environment,

---

[3] when the agent does not have an access to the true state, $s$, empowerment is calculated as the maximal mutual information between actuators and sensors given an estimated state.

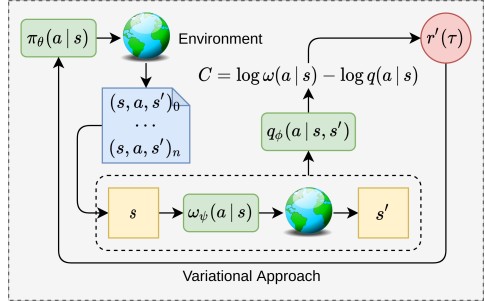

Figure 1: Estimation of empowerment via variational lower bounds.

$(s, a, s')$. The source policy $\omega_\psi(a \mid s)$ generates the action $a$ at every state $s$ visited by the controller policy, and the environment transits to the next state, $s'$ (yellow block). The inverse channel policy $q_\phi(a \mid s, s')$ is regressed to $a$ given $s$ and and $s'$, generating the intrinsic reward, $C = \log \omega_\psi(a \mid s) - \log q_\phi(a \mid s, s')$, which is used as the reward to improve the controller policy via reinforcement learning. This scheme aims to transfer the intrinsically motivated agent to the state of the maximal empowerment in the environment.

### 2.2.2 EMPOWERMENT FOR GAUSSIAN CHANNELS

The Gaussian linear channel in our case is given by:

$$O = G(s) \cdot A + \eta, \tag{4}$$

where $O$ and $A$ are the observation and actuator sequences with dimensions $d_O \times 1$ and $d_A \times 1$, respectively; $G(s)$ is a matrix with dimensions $d_O \times d_A$; and $\eta$ is multivariate Gaussian noise with dimension $d_O \times 1$. Following Salge et al. (2013), we assume the noise is distributed as a Gaussian. For completeness, Appendix A provides the analytical derivation of $G(s)$ in the case of known dynamics for pendulum for the action sequence $A$ with length 3. In contrast, the current work proposes a method how to estimate $G(s)$ for arbitrary dynamics and arbitrary length of the action sequence from samples. The details of the calculation of Gaussian channel capacity are included in Appendix B.

The key observation in our method is that we can learn $G(s)$ from samples, which allows us to estimate empowerment efficiently for arbitrary dynamics, including for high-dimensional dynamical systems or from images (pixels). As a result, it is not required to resort to lower bounds as done in prior work. In Section 4, we describe how to learn $G(s)$ from samples using deep neural networks.

## 3 RELATED WORK

In this section, we review the relevant prior work and state of the art in empowerment estimation, emphasizing the main differences between these existing works and our proposed approach.

**Estimation via Gaussian channel.** The representation given by Equation 4 was previously considered in (Salge et al. (2013)), where the channel $p(O \mid A, s)$ was assumed to be known. In our current work, we remove this assumption and extend the applicability of the efficient channel capacity solution given by Equation 11 to arbitrary observations, including images.

**Variational approaches to estimation.** The lower bound in Equation 2 is used in (Mohamed & Rezende (2015); Gregor et al. (2017); Karl et al. (2019)), where different additional assumptions are made. Specifically, Mohamed & Rezende (2015) estimates empowerment in open loop, while Gregor et al. (2017); Karl et al. (2019) provide a solution for closed-loop empowerment, and Karl et al. (2019) assumes differentiability of the dynamics. In our work, we do not assume differentiable dynamics and we do not resort to variational bounds, aiming to achieve a more stable, sample-efficient, and generalizable estimator of empowerment without losing the essential properties of the empowerment function, as explained in the Introduction. The works by (Mohamed & Rezende (2015); Gregor et al. (2017)) precede that by (Karl et al. (2019)), and we also experimentally find the latter to be more stable. Consequently, in Section 5 we provide comparisons to Karl et al. (2019), which represents the current start of the art in this line of research.

**Skill discovery.** An approximation of the lower bound in Equation 3 is successfully used by Sharma et al. (2020) in the context of learning a set of skills. Another relevant and important work is Eysenbach et al. (2019). Citing the authors, their work is about *"maximizing the empowerment of a hierarchical agent whose action space is the set of skills"*. Section 5 provides comparisons between our empowerment estimator and these methods with regards to stability in training, sample efficiency, and correctness of the empowerment landscape generated by both methods. Although they aim to achieve a different goal from our work – unsupervised skill discovery vs unsupervised stabilization – their objectives are the same type us ours: maximizing mutual information between actuators and sensors. In Section 5 we show that the empowerment landscape derived by these methods converge to the true empowerment landscape in simple cases, but fail to converge to a known empowerment landscape in more complicated dynamical systems.

These works represent the state of the art in empowerment estimation. We believe a thorough comparison between our method and these works, as provided in Section 5, clearly demonstrate the main properties of our method and clarify our contribution to the improvement of sample-based empowerment estimation in dynamical systems. As mentioned in the introduction, empowerment is useful for a broad spectrum of different applications. In this section, we highlighted the most relevant and updated methods, which we hope are helpful to see the merits and advances of our approach.

Our improved empowerment estimation possesses the essential properties of the empowerment function: in particular, it converges to the correct empowerment landscape. The training has improved stability. Furthermore, it allows for the estimation of the empowerment function from images, which we demonstrate in Section 5, where we show purely intrinsically motivated (empowerment-based) stabilization of inverted pendulum from images. Accurate estimation of empowerment from images significantly expands the possible breadth of applications for empowerment.

## 4 PROPOSED METHOD

The desiderata for our method as follows:

- to learn the channel matrix ($G(s)$ in Equation 4 from samples for arbitrary dynamics.
- to utilize general RL optimizers to reliably train intrinsically motivated policies.
- to enable empowerment maximization from arbitrary dynamics using latent representations.

These are desirable as: the ability to learn $G(s)$ from samples would allow us to estimate empowerment by the efficient line search algorithm in Equation 12, recent advances in RL algorithms can alleviate challenges in stability and sample efficiency, and latent representations would allow our method to be applicable even when the observation or underlying dynamics do not satisfy the Gaussian Linear Channel assumption.

In this work, we present Latent Gaussian Channel Empowerment, or Latent-GCE in short. Schematically, our method is depicted in Figure 2. The controller policy $\pi_\theta(a \mid s)$ collects trajectories in the environment, $(s, a, s')$, where $s$ is either a state or image observation. The channel matrix $G_\chi(s_t)$ predicts the final state of a trajectory corresponding to the action sequence $a_t^{T-1}$, starting from $s_t$. After performing singular value decomposition on $G_\chi(s_t)$, the empowerment value $\mathcal{E}(s_t)$ is estimated by Equation 11 and Equation 12. The empowerment along trajectories provide intrinsic rewards, denoted by $r'(\tau)$, for the update of the controller policy, $\pi_\theta$, using off-the-shelf RL algorithms. Our method is summarized in high level in Algorithm 1.

### 4.1 EMBEDDING OF OBSERVATIONS AND ACTION SEQUENCES

While the Gaussian Linear Channel defined in Equation 4 provides a good estimate of simple dynamic systems, we want our empowerment estimator to generalize to arbitrary dynamics, including high-dimensional systems, and environments where the observations are images. Using autoencoders, we embed the observations and action sequences such that their latent vector representations satisfy the Gaussian Linear Channel property. We use CNN encoder and decoder for image observations, or fully-connected networks for state observations. The encoder-decoder pair for action sequences are also parameterized by fully-connected networks. Let $f_\chi, f_\chi^{-1}, g_\chi, g_\chi^{-1}$ denote the encoder-decoder pairs for the observations and action sequences, $f_\chi$ maps the observations into $d_f$ dimensional latent

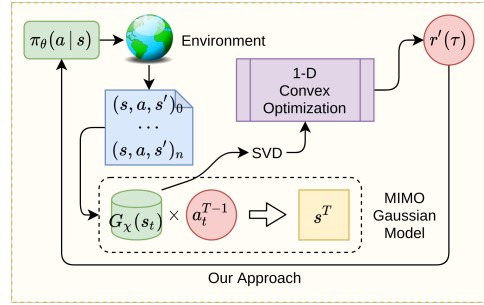

Figure 2: Empowerment maximization via Latent-GCE

---

**Algorithm 1** Latent Gaussian Channel Empowerment Maximization

---

1: **Input:** $\pi_{\theta_0}(a \mid s)$ - control policy, $H$ - prediction horizon.
2: **for** $i = 0$ **to** $N$ **do**
3:     $\{\tau_k\}_{k=1}^K \leftarrow$ run policy $\pi_{\theta_i}(a \mid s)$                       {data collection in environment}
4:     extract tuples $(s_t, a_t^{H-1}, s_{t+H})$ from $\{\tau_k\}_{k=1}^K$     {data arrangement for Gaussian channel}
5:     **for** $j = 1$ **to** $M$ **do**
6:         $\chi^* \leftarrow \arg\min_{\chi} \sum_{tuples} \left\| G_\chi(s_t) \cdot a_t^H - s_{t+H} \right\|_2^2$             {projection to Equation 4}
7:     **end for**
8:     $\{\sigma_i^*(s_t)\}_{i=1}^k \leftarrow \text{SVD}(G_{\chi^*}(s_t))$                {an alternative explained in Appendix C}
9:     $\mathcal{E}(s_t) \leftarrow \frac{1}{2} \sum_{i=1}^k \log(1 + \sigma_i^*(s)p_i^*)$             {Equation 11 and Equation 12}
10:    reinforce $\pi_{\theta_i}(a \mid s)$ with $r'(\tau) = \sum_{t=1}^T \gamma^{t-1} \mathcal{E}(s_t)$.           {policy update}
11: **end for**

---

states, and $g_\chi$ maps the $H$-step action sequences into $d_g$ dimensional latent actions. Our Latent Gaussian Linear Channel is formulated as follows:

$$f_\chi(s_{t+H}) = G_\chi(f_\chi(s_t)) \cdot g_\chi(a_t^H) + \eta \tag{5}$$

## 4.2 Learning Channel Matrix G

The channel matrix $G_\chi(f_\chi(s_t))$ has dimensions $d_f \times d_g$, where $d_f$ and $d_g$ are the dimensions of latent states and actions as defined in section 4.1. In the Latent Gaussian Linear Channel, as formulated in Equation 5, the ending latent state is approximated by a matrix-vector multiplication between the channel matrix $G_\chi(f_\chi(s_t))$ and the latent action $g_\chi(a_t^H)$. The mapping from a latent state to its channel matrix is parameterized by a 3-layer fully-connected neural network. The output layer of the network contains $d_f \cdot d_g$ neurons, which is then reshaped into the $d_f \times d_g$ matrix.

## 4.3 Overall Optimization Objectives

The overall training objective for our Latent-GCE estimation consists of two parts: reconstruction loss and latent prediction loss. The reconstruction loss $L_R$ is the average L2 distance between the decoded latent representations and the original encoder inputs. Minimizing $L_R$ encourages the autoencoders to learn meaningful latent representations. The prediction loss $L_P$ is the average L2 distance between the matrix-vector product and the true latent ending state. Minimizing $L_P$ forces the latent space to maintain the Gaussian Linear Channel structure:

$$
\begin{aligned}
L(\chi) &= L_R + L_P \\
&= \mathbb{E}_s \left\| f_\chi^{-1}(f_\chi(s)) - s \right\|_2^2 + \mathbb{E}_{a_t^H} \left\| g_\chi^{-1}(g_\chi(a_t^H)) - a_t^H \right\|_2^2 + \\
&\quad \mathbb{E}_{(s_t, a_t^H, s_{t+H})} \left\| G_\chi(f_\chi(s_t)) \cdot g_\chi(a_t^H) - f_\chi(s_{t+H}) \right\|_2^2
\end{aligned}
\tag{6}
$$

Overall, our Latent-GCE Maximization scheme (Figure 2) trains two sets of parameters, $(\theta, \chi)$, while VLB methods (Figure 1) generally needs three sets, $(\theta, \phi, \psi)$. The reduced trainable components

suggest that our method might be superior in terms of better stability and quicker convergence, which we confirm in our experiment section.

# 5 EXPERIMENTS

## 5.1 UNSUPERVISED STABILIZATION OF CLASSIC DYNAMIC SYSTEMS

As discussed in Section 1, a control policy that maximizes agent empowerment should lead to agent stabilization, even at an unstable equilibrium. To evaluate this, we compare our method with two prior works by Mohamed & Rezende (2015) and Karl et al. (2019) on their ability to stabilize simple dynamic systems. Specifically, in the 2D ball-in-box environment, empowerment is highest at the center of the box, where the ball is free to move in all directions; in the pendulum environment, empowerment is highest at the up-right position, where gravity assists in reaching other states. We compare average squared distance away from the stable position during policy training. As shown in Figure 3, all three methods are successful on the simple ball-in-box environment, whereas only our method consistently stabilizes the pendulum at an unstable equilibrium. We refer readers to Appendix D for detailed visualization of pendulum empowerment/policy learning using our method, and Appendix E for results on cart-pole.

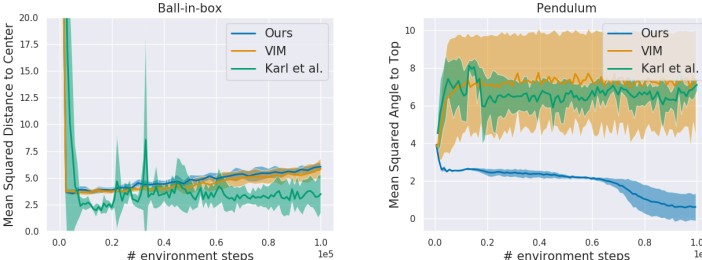

Figure 3: Evaluating whether methods learn to stabilize simple dynamical systems. All methods succeed for ball-in-box, whereas only our method succeeds for pendulum.

## 5.2 QUALITATIVE CONVERGENCE OF DIFFERENT ESTIMATORS

We now qualitatively compare the convergence of different empowerment estimators to the true empowerment function. Not only do we include solutions by VLB methods that approximate the classic empowerment definition (Mohamed & Rezende (2015); Karl et al. (2019)), but we also compare with recent works that employ empowerment-like objectives for skill discovery (Eysenbach et al. (2019); Sharma et al. (2020)). From Figure 4, we observe that in simple environments such as ball-in-box, all estimators converge to the correct landscape with high empowerment values in the middle and low values near the boundaries, corresponding to previous work (Klyubin et al. (2005)). For the pendulum environment, it is clear that while both the method by Karl et al. (2019) and ours succeed in creating results similar to the analytical solution (the ground truth), ours outputs an empowerment landscape that is smoother and more discerning. Detailed derivation of the analytical solution is included in Appendix A.

## 5.3 STABILITY OF EMPOWERMENT ESTIMATIONS

Besides the convergence to correct empowerment values, the stability and sample efficiency are also key aspects to determine whether the empowerment estimators are suitable as reliable intrinsic reward signals. We compute the variance of average empowerment values over 10 seeds as a measurement of algorithm stability. Since the outputs by existing methods are on different scales, we instead compare the relative standard deviation. As shown in Figure 5, our method outperforms the existing sample-based methods in that it has lower variance in training and faster convergence to the final estimator. These properties make our method competent as a reliable unsupervised training signal.

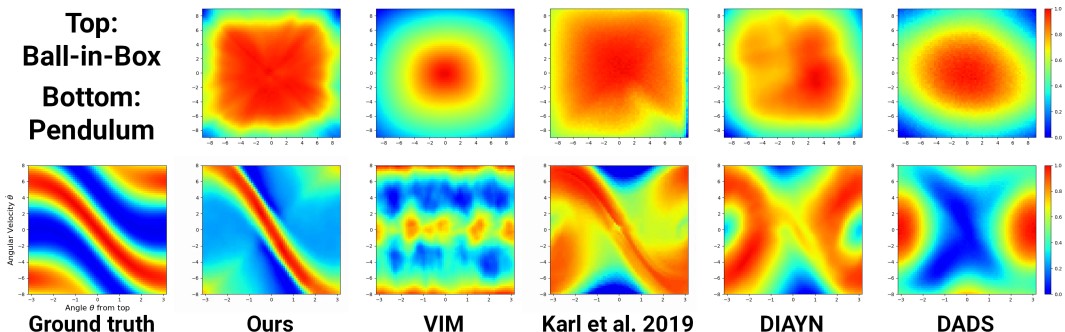

Figure 4: Empowerment landscapes (normalized) for ball-in-box (top) and pendulum (bottom). In ball-in-box, central positions correspond to high empowerment. For pendulum, upright positions correspond to high empowerment; x-axis is the angle to top, y-axis is the angular velocity. From left to right, the plots show: ours, Mohamed & Rezende (2015), Karl et al. (2019), Eysenbach et al. (2019), Sharma et al. (2020).

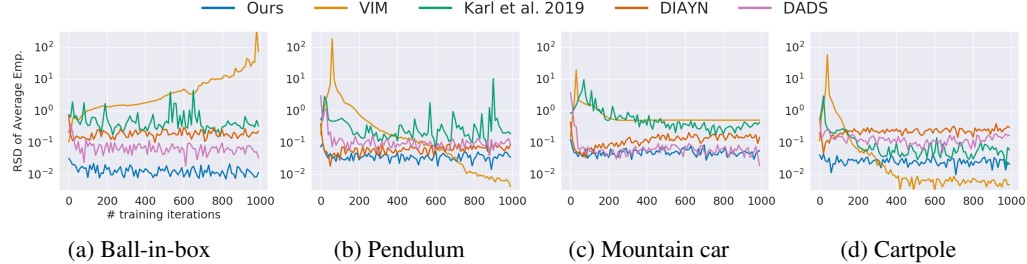

Figure 5: Relative standard deviation (RSD) in average empowerment value across 10 seeds. Our method converges faster to a lower RSD during training.

## 5.4 EMPOWERMENT ESTIMATION IN HIGH-DIMENSIONAL SYSTEMS

We verify the applicability of Latent-GCE in high-dimensional systems from two aspects: first, the latent space should be invertible - able to predict the true underlying dynamics; moreover, the empowerment maximization policy should develop stabilizing behaviors.

### 5.4.1 PIXEL-BASED PENDULUM

We use a pendulum environment that supplies $64 \times 64$ images as observations. As input to our Latent-GCE algorithm, we concatenate two consecutive frames so as to capture the angular velocity. In Figure 6, we start the pendulum at its up-right position $s_0$. Using the encoder $f_\chi$, we map the images into the latent representation $f_\chi(s_0)$. Given different action sequences $a_0^H$, we predict the ending latent state as $z_{t+H} = G_\chi(f_\chi(s_t)) \cdot g_\chi(a_t^H)$. If our learned Latent-GCE model captures the true system dynamics, we expect the reconstruction of the ending state $f_\chi^{-1}(z_{t+H})$ to match the ground truth $s_{t+H}$. This property holds true as verified in Figure 6. More importantly, as shown in Figure 7, the empowerment estimations given by our method is not only close to the analytical solution, but also trains a policy that swing the pendulum up from bottom to balance at the top. On the other hand, VIM by Mohamed & Rezende (2015) which we compare as a baseline, fails to assign a high empowerment estimation for the up-right position.

### 5.4.2 HOPPER

We evaluate the ability of Latent-GCE to stabilize the Hopper environment from OpenAI Gym. This environment is more challenging because of higher state dimension and long horizon (64) for empowerment estimation. Here we expect an empowerment maximization policy to balance the hopper at its standing position where it is able to move left and right freely. In Figure 8 we see our method successfully balances hopper whereas a random policy inevitably makes the hopper fall over.

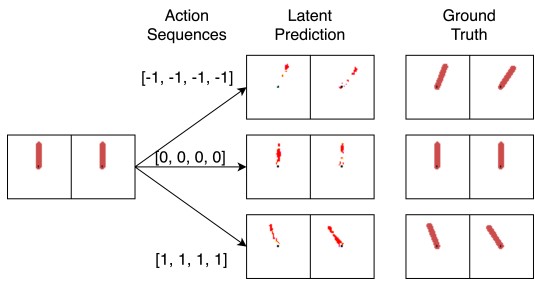

Figure 6: Reconstruction in the latent dynamics vs the ground truth. Starting from the same upright position, a different action sequence is taken in each row (action +1: counter-clockwise torque, 0: no torque, -1: clockwise torque). Reconstruction matches the actual observation in all cases.

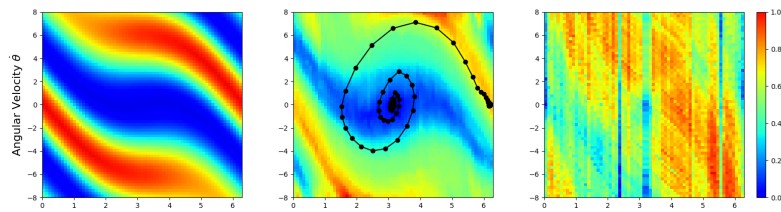

Figure 7: Empowerment estimates for pixel-based pendulum. Latent-GCE learns a reasonable empowerment function from pixels. Left: Analytical (x axis shifted by $\pi$ from Figure 4). Middle: Latent-GCE along with the swinging-up trajectory of the final control policy. Right: VIM.

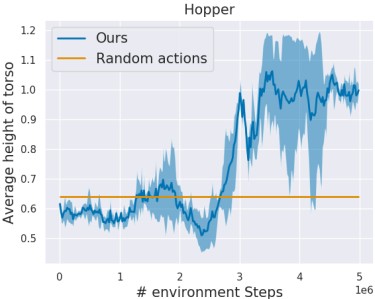

Figure 8: Average torso height over an evaluation episode of length 200 for the Hopper environment. Our method stabilizes the hopper to a standing position ($z \approx 1$).

## 6 SUMMARY AND DISCUSSION

In this work we introduce a new sample-based method for empowerment estimation in dynamical systems. The key component of our method is the representation of dynamical system by the Gaussian channel, which allows us to estimate empowerment by convex optimization. Our estimator converges to true empowerment landscape both from raw states and images. Empowerment is used in various applications and our improved estimator has a potential to improve these applications. For example in Appendix F, we show that when the agent is trained with an empowerment-aware objective, it develops distinguishable levels of attention to safety. Moreover, a natural continuation of this work is to learn skills using our representation of empowerment by Gaussian channel, rather than by VLB.

ACKNOWLEDGMENTS

This research is supported by Berkeley Deep Drive, Intel, and Amazon through providing AWS cloud computing services.

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

## A    ANALYTICAL DERIVATION OF $G(s)$ FOR INVERTED PENDULUM

The main steps in the derivation of $G(s)$ appear below. The full derivation appears in Section 4.1 at Salge et al. (2013). The current state of the pendulum, $s_t$, is given by $\theta_t$ and $\dot{\theta}_t$:

$$s_t = \begin{bmatrix} \theta_t \\ \dot{\theta}_t \end{bmatrix}, \tag{7}$$

where $\theta$ is measured from the upright position. We consider the development of the pendulum for three time steps of duration $dt$. The behaviour of the pendulum can be approximate linearly at every time step for small enough $dt$. Then, the next state of the pendulum is given by:

$$s_{t+dt} = s_t + A_t dt + B_t a_t, \tag{8}$$

where $a_t$ is the agent's action, $A_t = \begin{bmatrix} \dot{\theta}_t \\ \frac{g}{l}\sin(\theta_t) \end{bmatrix}$, and $B_t = \begin{bmatrix} 0 \\ 1 \end{bmatrix}$. For the next time step we need to calculate a new matrix, $A$, because the non-linearity of the system causes $A_{t+dt}$ to depend on the new state. Consequently, $A_{t+dt} = \begin{bmatrix} \dot{\theta}_{t+dt} \\ \frac{g}{l}\sin(\theta_{t+dt}) \end{bmatrix}$. A similar matrix can be calculated for $A_{t+2dt}$. An propagation of pendulum dynamics for 3 time steps allows us to calculate $s_{t+3dt}$ explicitly, which appears as:

$$\theta_{t+3dt} = dt(\frac{gdt}{l}\sin(dt\dot{\theta}_t + \theta_t) + \frac{gdt}{l}\sin(\theta_t) + \dot{\theta}_t)$$
$$+ dt(\frac{gdt}{l}\sin(\theta_t) + 2\dot{\theta}_t) + \theta_t + 3dta_t + 3dta_{t+dt},$$

and,

$$\dot{\theta}_{t+3dt} = \frac{gdt}{l}\sin(dt\dot{\theta}_t + \theta_t) + \frac{gdt}{l}\sin(\theta_t) + \dot{\theta}_t$$
$$+ \frac{gdt}{l}\sin(dt(\frac{gdt}{l}\sin(\theta_t) + 2\dot{\theta}_t) + \theta_t + 2dta_t) + dta_t + dta_{t+dt} + dta_{t+2dt},$$

where $a_{t+dt}$ is the action at the next step. Our goal is to represent $s_{t+3dt}$ by:

$$s_{t+3dt} = K + G(s_t) \cdot \begin{bmatrix} a_t \\ a_{t+dt} \\ a_{t+2dt} \end{bmatrix}, \tag{9}$$

which is achieved by linearisation $\theta_{t+3dt}$ and $\dot{\theta}_{t+3dt}$ around $a_t = 0$. A straightforward calculation shows that the matrix $G(s_t)$ is given by:

$$G(s_t) = \begin{bmatrix} 2dt & dt & 0 \\ \frac{dt^2 g}{l}\cos(dt(\frac{gdt}{l}\sin(\theta_t) + 2\dot{\theta}_t) + \theta_t) + 1 & 1 & 1 \end{bmatrix}. \tag{10}$$

This calculation can be continued beyond 3-steps, involving more complicated terms in the expansions. Given $G(s_t)$ in Equation 10 one can represent $s_{t+3dt}$ by Equation 9, where the matrix $K$ does not affect the mutual information Cover & Thomas (2012). Adding to Equation 9 the normal noise, $\eta$, as suggested in Salge et al. (2013), we get the representation in Equation 4. In contrast, we propose to learn $G(s)$ from samples via interaction with the environment for an arbitrary dynamics and an arbitrary number of steps (c.f. 3 steps in the above derivation). Using $G(s)$ from Equation 10, we can calculate the analytical (ground truth empowerment landscape), which is shown at Figure 9

## B    GAUSSIAN CHANNEL CAPACITY

The capacity of the channel in Equation 4 is given by, (Cover & Thomas (2012)):

$$\mathcal{E}(s) = \max_{\substack{p_i \geq 0 \\ \sum_i^k p_i \leq P}} \frac{1}{2}\sum_{i=1}^{k}\log(1 + \sigma_i(s)p_i) \tag{11}$$

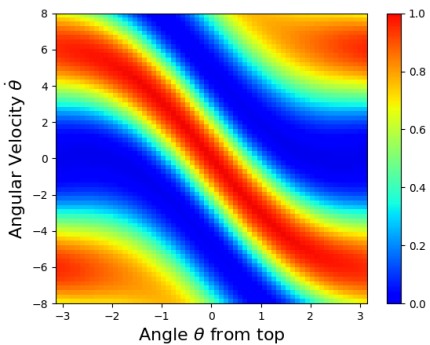

Figure 9: Analytical Empowerment Landscape

where $\sigma_i(s)$, $p_i$, $k$, and $P$, are the i-th singular value of $G(s)$, the corresponding actuator signal power, the number of nonzero singular values, and the total power of the actuator signal, respectively. The optimal solution to Equation 11 is given by:

$$p_i^*(\nu) = \max(1/\nu - 1/\sigma_i(s), 0), \quad \sum_i^k p_i^*(\nu) = P, \tag{12}$$

where $\nu$ is found by a line search (Cover & Thomas (2012)) to satisfy the total power constraint.

## C  SVD ALTERNATIVE

The singular values, $\sigma_i(s_t)$, can be estimated directly without the singular values decomposition by representing the channel matrix as $G_\chi(s_t) = U_\chi(s_t) * Diagm(\Sigma_\chi(s_t)) * V_\chi(s_t)$ with two additional orthonormality constraints: $\left\|U_\chi(s_t)U_\chi^\dagger(s_t) - \mathbf{I}_{\{d_s \times d_s\}}\right\|_2^2$ and $\left\|V_\chi(s_t)V_\chi^\dagger(s_t) - \mathbf{I}_{\{d_a(H-1) \times d_a(H-1)\}}\right\|_2^2$. In our experiments, we did not experience a bottleneck in calculating the singular values decomposition for both raw states and images. While we don't implement this in our experiments, this alternative direct calculation can be applied when SVD is undesired.

## D  TWO PHASES IN EMPOWERMENT-BASED STABILIZATION

Empowerment of the pendulum is highest when it is stationary at the up-right position. In this section, we show the evolution of the empowerment landscape, the corresponding control policy and the state distribution as we run Algorithm 1 without any external reward. In Figure 10, we observe that Algorithm 1 includes two phases: empowerment learning and policy learning. Empowerment converges from epoch 0 to 10 and then, policy converges from epoch 10 to 50.

## E  CART-POLE SWING-UP

The cart-pole environment is made up of a stiff pole with one end fixed to a pivot on a cart movable along one axis. The cart can be pushed left or right within a confined region. The state space of cart-pole has 4 components: cart position, cart velocity, pole angle, pole angular velocity. We expect high empowerment when the pole is up right, because starting there, gravity is able to help the system reach a broader set of future states. To measure whether Algorithm 1 successfully brings the cart-pole to up-right position, we plot the average squared angle to the top against the number of training steps. We also include 2 different reward functions that directly aim this objective for comparison.

As shown in Figure 11, the empowerment reward, without the need of prior knowledge of the environment, is able to achieve similar final performance compared to direct reward signals. Moreover, it learns this objective at a speed on par with the dense reward.

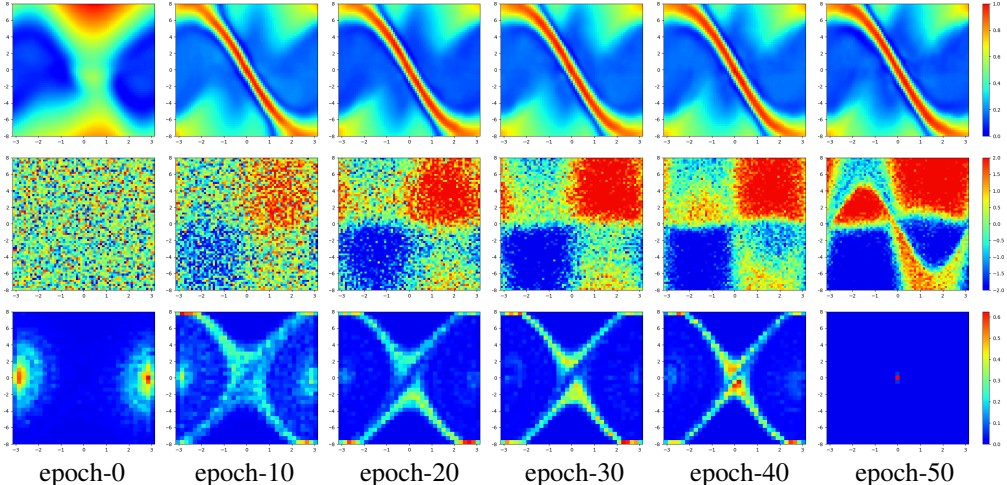

epoch-0 epoch-10 epoch-20 epoch-30 epoch-40 epoch-50

Figure 10: Snapshots of the empowerment landscape (top), the control policy (middle) and the state concentration (bottom). The x-axis is $\theta$ (angle) and the y-axis is $\dot{\theta}$ (angular velocity). The angle is measured from the upright position, ($\theta = 0\,\mathrm{rad}$ corresponds to the upright position). Each epoch corresponds to $2 \cdot 10^4$ steps. The policy corresponds to the optimal policy for the stabilization task with external reward (Doya (2000)).

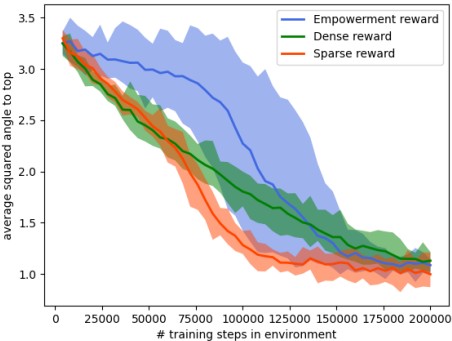

Figure 11: Cart-Pole - Comparison of unsupervised and supervised reward functions. The "dense reward" is $-\theta^2$ and the "sparse reward" is $\mathbb{1}\{|\theta| < \frac{\pi}{10}\}$.

## F    EMPOWERMENT ESTIMATION FOR SAFETY

A useful application of empowerment is its implication of safety for the artificial agent. A state is intrinsically safe for an agent when the agent has a high diversity of future states, achievable by its actions. This is because in such states, the agent can take effective actions to prevent undesirable futures. In this context, the higher its empowerment value, the safer the agent is. In this experiment, we first check that our calculation of empowerment matches the specific design of the environment. Additionally, we show that empowerment augmented reward function can affect the agent's preference between a shorter but more dangerous path and a longer but safer one.

**Environment:** a double tunnel environment implemented with to the OpenAI Gym API (Brockman et al., 2016). Agent (blue) is modeled as a ball of radius 1 inside a 20×20 box. The box is separated by a thick wall (gray) into top and bottom section. Two tunnels connect the top and bottom of the box. The tunnel in middle is narrower but closer to the goal compared to the one on the right.

**Control:** In each time step, the agent can move at most 0.5 unit length in each of x and y direction. If an action takes the agent into the walls, it will be shifted back out to the nearest surface.

**Reset criterion:** each episode has a maximum length of 200 steps. The environment resets when time runs out or when the agent reaches the goal.

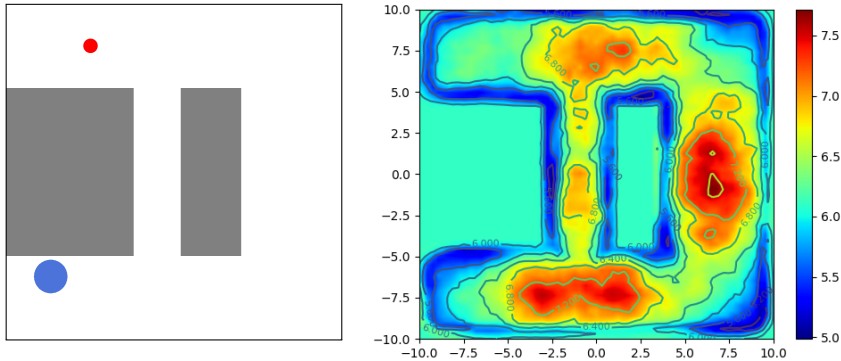

Figure 12: **Left sub-figure**: Double tunnel environment. The goal is marked in red. The agent in blue. **Right sub-figure**: Empowerment landscape for the tunnel environment. The values of empowerment reduce at the corner and inside the tunnel where the control of the agent is less effective compared to more open locations.

Since the tunnel in the middle is narrower, the agent is relatively less safe there. The effectiveness of the control of the agent is damped in 2 ways:

1. In a particular time step, it's more likely for the agent to bump into the walls. When this happens, the agent is unable to proceed as far as desired.

2. The tunnel is 10 units in length. When the agent is around the middle, it will still be inside the tunnel in the 5-step horizon. Thus, the agent has fewer possible future states.

We trained the agent with PPO algorithm (Schulman et al., 2017) from OpenAI baselines (Dhariwal et al., 2017) using an empowerment augmented reward function. After a parameter search, we used discount factor $\gamma = 0.95$ over a total of $10^6$ steps. The reward function that we choose is:

$$R(s, a) = \mathbf{1}_{goal} + \beta \times Emp(s)$$

where $\beta$ balances the relative weight between the goal conditioned reward and the intrinsic safety reward. With high $\beta$, we expect the agent to learn a more conservative policy.

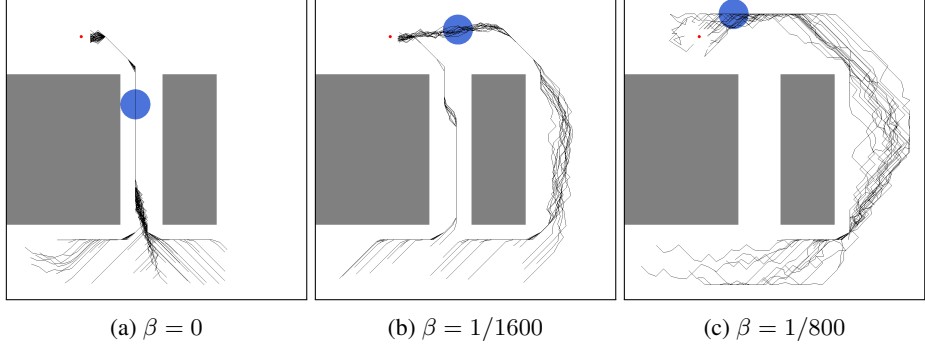

(a) $\beta = 0$        (b) $\beta = 1/1600$        (c) $\beta = 1/800$

Figure 13: Trajectories of trained policy. As $\beta$ increase, the agent develops a stronger preference over a safer route, sacrificing hitting time.

The results from the tunnel environment again support our proposed scheme. First, the empowerment landscape matches our design of the environment. Second, high quality empowerment reward successfully alters the behavior of the agent.

# G  Experiment Details

## G.1  Use of neural encoders

In our state-based classic control environments, the encoder/decoder pairs for observations and actions are set to be the identity map. In this case, the sequences of actions are simply concatenated. For the Hopper environment and pixel-based pendulum, encoders and decoders are parameterized by neural networks.

## G.2  Details on neural network layers

### G.2.1  CNN for image encoding

(h1) 2D convolution: 4 filters, stride 2, 32 channels, ReLU activation
(h2) 2D convolution: 4 filters, stride 2, 64 channels, ReLU activation
(h3) 2D convolution: 4 filters, stride 2, 128 channel, ReLU activations
(h4) 2D convolution: 4 filters, stride 2, 256 channel, ReLU activations
(out) Flatten each sample to a 1D tensor

### G.2.2  Deconvolutional net for image reconstruction

(h1) Fully connected layer with 1024 neurons, ReLU activation
(h1') Reshape to $1 \times 1$ images with 1024 channels
(h2) 2D conv-transpose: 5 filters, stride 2, 128 channels, ReLU activation
(h3) 2D convolution: 5 filters, stride 2, 64 channels, ReLU activation
(h4) 2D convolution: 6 filters, stride 2, 32 channel, ReLU activations
(out) 2D convolution: 6 filters, stride 2, $C$ channel

### G.2.3  MLP for action sequence encoding

(h1) Fully connected layer with 512 neurons, ReLU activation
(h2) Fully connected layer with 512 neurons, ReLU activation
(h3) Fully connected layer with 512 neurons, ReLU activation
(out) Fully conected layer with 32 output neurons

### G.2.4  MLP for action sequence reconstruction

(h1) Fully connected layer with 512 neurons, ReLU activation
(h2) Fully connected layer with 512 neurons, ReLU activation
(out) Fully conected layer with $T \times d_a$ neurons, tanh activation then scaled to action space

### G.2.5  MLP for transition matrix A

(h1) Fully connected layer with 512 neurons, ReLU activation
(h2) Fully connected layer with 512 neurons, ReLU activation
(h2) Fully connected layer with 512 neurons, ReLU activation
(out) Fully conected layer with $d_f \times d_g$ neurons

## G.3  Dynamics-Aware Discovery of Skills (DADS)

For our experiments using DADS, we use policy and discriminator architecture of two hidden layers of size 256 with ReLU activations and a latent dimension of 2. Our discount factor is .99. As per suggestions from the original paper, we use 512 prior samples to approximate the intrinsic reward, run 32 discriminator updates and 128 policy updates per 1000 timesteps; learning rates of $3 \times 10^{-4}$ and batch size of 256 are used throughout. We use a replay buffer to help stabilize training of size 20000, used only for policy updates.

### G.4 HOPPER EXPERIMENT DETAILS

Each episode is set to a fixed length of 200 steps. We use PPO as the RL backbone for our control policy. The policy is updated every 1000 steps in the environment, with learning rate of 1e-4 and $\gamma = 0.5$.

