# OpenReview forum: "Efficient Empowerment Estimation for Unsupervised Stabilization"
_ICLR.cc/2021/Conference — ICLR 2021 Poster_

### Official Review · AnonReviewer2 · 2020-10-26
**Almost closed form empowerment estimation, but the assumptions are unrealistic**

**Rating:** 5
**Confidence:** 4

**Review:**

Summary

The paper studies reward-free reinforcement learning (RL) methods based on empowerment. The authors propose a technique for empowerment estimation under the assumption that a state after H timesteps can be factorized as a product of the current action and a matrix G(s) that depends on the current state. In contrast to the existing methods that rely on the optimization of variational lower bounds on mutual information for empowerment estimation, the technique allows having an almost closed-form expression for empowerment. The authors qualitatively demonstrate the convergence of their method to the true empowerment function on simple environments such as 2D ball-in-box as well on image-based Pendulum environment.

Strengths
- Under the proposed factorization, the empowerment estimation requires only calculating SVD of the G(s) matrix and line search for satisfying constraints.
- The plots for 2D ball-in-box environment demonstrate that the proposed method estimates empowerment more accurately than other algorithms that have empowerment estimation as a component.

Weaknesses
- The experimental results could be demonstrated on harder environments such as MuJoCo. For example, DIAYN [1], one of the methods the authors compare with, provides results on Ant and Cheetah environments.
- The proposed factorization, which is a key component for empowerment estimation, might be too restrictive and not scalable. Moreover, the reviewer did not find the details on how exactly G(s) is calculated except that it is the output of a neural network.
- The positioning of the paper is unclear. If the main contribution is the improvement over existing reward-free RL methods, then it is more appropriate to demonstrate that the agent that uses estimated empowerment achieves high returns in the environment or has an interesting emergent behavior. If the main contribution is the improvement in empowerment estimation, it is more appropriate to compare with methods designed for mutual information estimation.
- The overall clarity of the paper could be significantly improved.

Recommendation

The reviewer votes for rejection. The methods the authors compare with are not designed for empowerment estimation, they use empowerment only as a proxy reward e.g. for overcoming exploration or learning skills. Moreover, it is unclear whether the method will scale beyond simple environments. Addressing the outlined weaknesses might increase the final score.

**

Post-rebuttal update: the score is increased from 3 to 5. See the response to authors' comments.

**

Notes:
1. The connection between equations (1) and (3) which both define empowerment is unclear.
2. It would be helpful to provide more experimental details. For example, the authors state that their method requires only training only two neural networks compared to three networks for variational lower bound methods. However, the benefit of this is unclear without providing the architectures of the neural networks and the total number of parameters.
3. The code and additional materials were not available in the Google Drive folder indicated in Supplementary Material at the moment of submitting the review. However, this did not influence the final score.

[1] Eysenbach, Benjamin, Abhishek Gupta, Julian Ibarz, and Sergey Levine. "Diversity is all you need: Learning skills without a reward function." arXiv preprint arXiv:1802.06070 (2018).

---

> ### Author Response · Authors · 2020-11-23
> **Revision posted, including improved clarity in explanation, more detailed experiment/algorithm settings, and a re-organized suite of experiments for more appropriate positioning.**
>
> We thank the reviewer for acknowledging the advantage of our proposed method as an almost close form empowerment estimation. Meantime, we agree that the weaknesses pointed out are present in our original manuscript, as mentioned by other reviewers too. We’ve made a substantial revision to the paper which addresses all the issues, with emphasis on the clarity of the explanation of our method, and proper choice of baselines for comparison.
>
> 1. “The experimental results could be demonstrated on harder environments” -- While we had shown empowerment estimation on the pixel-based pendulum, which is harder than those in relevant works, it would indeed be more impressive if the method can be shown on other dynamic systems. In the revision, we added unsupervised stabilization of the MuJoCo Hopper, which uses significantly more complicated underlying dynamics and a much longer horizon for empowerment estimation. We favor the Hopper over the Ant and HalfCheetah to show a more impressive result because Ant and HalfCheetah remain standing even when small random actions are applied, while Hopper easily falls to the ground.
>
> 2. “The proposed factorization, which is a key component for empowerment estimation, might be too restrictive and not scalable.” -- Our insufficient explanation in the original manuscript must have caused this misunderstanding. A key contribution of our presented work is the ability to encode observations and action sequences to a latent Gaussian Channel. This ensures that our method remains applicable to pixel-based observations, long horizons, and arbitrary dynamics. In the revision, we make sure that this idea is clearly presented.
>
> 3. “The positioning of the paper is unclear.” -- This ties closely to the discussion by Reviewer 3. As a new algorithm for empowerment estimation, it makes more sense for our method to be compared with existing VLB empowerment estimators. In the revision, we’ve re-organized the experiment section to feature comparison with VIM (Mohamed & Rezende (2015)) and the work by Karl et al.. Relevant works on skill discovery are kept as additional comparisons.
>
> 4. “The connection between equations (1) and (3) which both define empowerment is unclear.” The definition of empowerment is provided by equation 1. Equation 3 shows the variational lower bound on the summand of Eq.1, where the variational distribution q(O|A, s) is introduced to approximate the true distribution p(O|A, s). The l.h.s of Eq.3 equals to the l.h.s. of Eq.1 without the maximum, which follows from the symmetry of mutual information: I[O; A | s] = I[A; O| s]. Eq.3 is provided for completeness of the presentation of prior work; specifically, the variational lower bound in Eq.3 is the objective in DADS, which we use in our benchmark.
>
> 5. “It would be helpful to provide more experimental details.” -- We agree that while experiment results show superior convergence, supplying details on neural network architecture can better substantiate the benefit of our method. In the revision, we’ve added more explanation of the method, training objectives, and exact NN layers used in our model.
>
> 6. “The code and additional materials were not available.” -- We apologize for the delay in code release. The code for our method will be made publicly available for the reproduction of our results.
>
> We thank the reviewer again for the constructive feedback which helps shape this revision. We hope the revision positions our work appropriately among relevant works and puts the paper in better shape for publication.

---

> > ### Comment · AnonReviewer2 · 2020-11-24
> > **Response to authors' comments**
> >
> > We thank the authors for carefully addressing the weaknesses outlined in the review.
> >
> > The results for Hopper make the claims in the paper more convincing. However, given the instability of the training trajectory, it is unclear whether the method will be consistently resulting in a high torso height given more training steps or will be fluctuating around the number achieved by a random policy.
> >
> > The other points in the response make sense. We update the score to 5 and encourage the authors to further improve the paper.
> >
> > Notes:
> > * Plot 5 would be more readable if the log scale of the y axis was used.

---

> > > ### Author Response · Authors · 2020-11-24
> > > **Hopper experiment extended to demonstrate convergence, Figure 5 re-plotted using log scale.**
> > >
> > > We thank the reviewer for acknowledging the positive changes we have made to the paper. We are genuinely happy that our major revision with more explanations and experiments properly addresses fellow reviewers' feedbacks and has made a difference.
> > >
> > > We've just uploaded a minor update to address the latest suggestions:
> > >
> > > 1. We re-ran our Hopper experiment for a longer duration and are able to verify that the final policies indeed converge to high torso height. Figure 8 in the paper is now replaced by the long version. We appreciate the elaborate review which helps make our message more convincing.
> > >
> > > 2. Thanks for the suggestion to plot Figure 5 on log scale! We have updated this figure in the latest revision, and indeed it makes the comparison much more readable.

---

### Official Review · AnonReviewer3 · 2020-10-29
**Interesting work in light of the revisions.**

**Rating:** 7
**Confidence:** 5

**Review:**

EDIT: the authors have removed the claims I was concerned by, as well as adding a requested baseline. The new quantitative results on stabilization also make their method much more compelling. The new hopper results are a bit noisy (would it stay stabilized if training continued?), but are still quite promising in that they show an ability to handle more complex dynamics.
4-->7, thanks for the great revision effort!

This work tackles the generally intractable problem of calculating channel capacity by learning a dynamics model with a highly constrained latent space (linear with Gaussian noise) that permits an efficient solution. This allows for using the exact empowerment (of the approximate dynamics) as the reward function for a reinforcement learning agent. This method is demonstrated on a few simple control problems, with the most difficult between a pendulum from pixel observations, which results in unsupervised stabilization.

The fundamental idea is sound; generalizing a special-case algorithm by forcing its constraints to hold in latent space has a long line of success (e.g. "Embed to Control"). And the model-based approach used here might allow for completely off-policy learning of empowerment. DADS is also model-based and could be adapted to optimize empowerment, but since its not a true lower-bound the case in favor of your approach over variational methods would be quite strong.

But the experiment baselines are not what they should be. Neither DADS nor DIAYN are measuring the same quantity as this method, namely empowerment: I(A^k; s_{t+k} | s_t). DADS is (almost) a lower bound on I(z ; s' | s) and DIAYN lower bounds I(z; s)+H(a|s, z). If there were some performance metric they all aim to achieve then this might be fine, but the only comparisons are to the quality of empowerment estimation (which they aren't estimating). I'd suggest switching to VIM[1] and VIC which lower bound I(A^k; s_{t+k} | s_t) and I(z; s_{t+k} | s_t) respectively.

A larger problem that is more easily remedied is the use of overstated claims:

"our method has a lower sample complexity" -- this is not shown. Yes, the empowerment of the modelled channel can be computed without additional interaction, but is building the model more sample efficient than getting a model-free estimate using a variational lower-bound? Maybe, but this isn't demonstrated.

"allows, for the first time, to estimate empowerment from images" -- this can only be true in the most contrived sense. While it's true that DIAYN and DADS both used explicit state, only DADS really exploits that decision architecturally. Indeed [2] includes a pixel-based version of DIAYN as a baseline without comment, and while that version of DIAYN doesn't perform that well, the method introduced there (VISR) tackles full Atari games, which constitute a much harder perceptual challenge than a pendulum on a blank background. Now, you could argue that none of these methods actually lower-bound empowerment (indeed, I just did in the previous paragraph). But adapting them lower-bound empowerment requires trivial changes e.g. for VISR you'd just have to evaluate the reverse predictor at the end of a trajectory rather than at every state. But one can look back far earlier. VIM [1] calculated empowerment from pixels in 2015. You could argue the pendulum environment is more complicated than the toy navigation tasks they used, but its hard to argue that it wouldn't work for pendulum since you didn't try it out.

I am not willing to see this work published in its current state, but I promise I'll support publication if 1) VIM is added as a baseline for all (including pendulum from pixels) experiments 2) the overstated claim are dropped.

[1] Mohamed, S., & Rezende, D. J. (2015). Variational information maximisation for intrinsically motivated reinforcement learning. In Advances in neural information processing systems (pp. 2125-2133).
[2] Hansen, S., Dabney, W., Barreto, A., Van de Wiele, T., Warde-Farley, D., & Mnih, V. (2019). Fast task inference with variational intrinsic successor features. arXiv preprint arXiv:1906.05030.

---

> ### Author Response · Authors · 2020-11-23
> **Revision posted with significantly improved clarity. VIM added as the main baseline for comparison.**
>
> We thank the reviewer for the detailed discussion on the placement of our work relative to the field. Especially, we agree that comparison with VIM/VIC is a necessity to position this work properly.
>
> 1. “The experiment baselines are not what they should be.” -- in this revision, we’ve added VIM as a baseline across the board when applicable. As the other baseline for direct comparison between our method and existing empowerment estimation, we supply differentiable dynamics to the method by Karl et al., which also computes the same VLB. We rearrange the experiment section to emphasize the comparison with these two methods while keeping the line of skill discovery works as additional comparisons.
>
> 2. “The most difficult between a pendulum from pixel observations.” -- besides the pixel-based pendulum result, we added results on Hopper which is also hard because of its more complicated underlying dynamics and the long horizon (64) for empowerment reasoning.
>
> 3. “And the model-based approach used here might allow for completely off-policy learning of empowerment.” -- Great observation! Indeed the samples used to train the encoding and transition matrix G don’t need to come from any specific distribution. We consider it a strength of our method that even random interactions with the environment can be used.
>
> 4. “A larger problem that is more easily remedied is the use of overstated claims” -- we carefully examined our claims in the initial submission. In this revision, we removed overclaims and added support for our contributions. We consider our method efficient due to its earlier convergence to its final empowerment estimation in terms of the number of interactions with the environment, as illustrated in the relative standard deviation comparison in Figure 5.
>
> Overall, we greatly improved the clarity of the paper and employed a more appropriate set of experiments in this revision. We hope that the changes make this work in better shape for publication.

---

> > ### Comment · AnonReviewer3 · 2020-11-24
> > **Thank you for the very thoughtful response.**
> >
> > I've updated my review to reflect your changes, and indeed I think this work definitely passes the bar for publication in its current state.

---

### Official Review · AnonReviewer1 · 2020-10-29
**Great work, unsure about the impact and relevance**

**Rating:** 6
**Confidence:** 2

**Review:**

First of all, the paper is well organized, provides a very clear description of the method.

Reproducing results
Will code be available? It is not mentioned that code is made available.

Correctness
Authors mention that training with empowerment converges to an unstable equilibrium (3rd paragraph in Introduction). This is not always the case but depends on the system.
The agent does not maximize the MI, but maximizes the potential MI, as Empowerment is not MI but channel capacity (4th paragraph Introduction).

Novelty/Relevance
The relevance is that they use a representation for the dynamical system (i.e. Gaussian channel), that should improve the quality of the Empowerment estimation.
They mention that they for the first time estimate empowerment from images. Karl et al. 2017 also use a DVBF, which also allows a transition in latent space from images observed.

Unclear
They estimate empowerment for a state s_t, which now is used to train a policy. However, normally the reward is a value for s_t, a_t pairs, so essentially a function of s_t+1. Should empowerment also be computed for s_t+1?

Small notes
Maximilian et al. should be Karl et al

I am uncertain whether the Gaussian channel is a contribution to the research community. Is it only useful for research done on empowerment? Can it be used for other systems that use a VLB?

---

> ### Author Response · Authors · 2020-11-23
> **Discussion about concurrent work, influence and potential future directions.**
>
> We appreciate that the reviewer finds our paper well organized and clear. Below we address each and every comment and suggestion.
>
> 1. “Will code be available?” -- Yes, and we apologize for the delay. The code will be made available publically for sake of reproducibility of the results.
>
> 2. “The agent does not maximize the MI, but maximizes the potential MI, as Empowerment is not MI but channel capacity” -- Indeed mutual information does not reflect the potential to achieve distinguishably a large number of future states by actions. In this work, we follow the classical works on empowerment by the group of Prof. Daniel Polani. Specifically, the work by “Jung et al. (2011)”, referred to by our current paper, demonstrates the stabilizing property of empowerment on a broad range of dynamical systems. In our current work, we scale up this method towards an efficient sample-based estimation of empowerment in high-dimensional settings, including image observations.  We explore a rigorous connection between empowerment (channel capacity between actions and states) and intrinsic properties of dynamical systems, (such as Lyapunov exponents), in our concurrent work, which is beyond the scope of the current paper. The current paper is focused on the new method for empowerment estimation.
>
> 3. Indeed, “Karl et al. 2017 also uses a DVBF” and suggests modeling dynamics in latent space. In our experiment, we helped their method by directly supplying a ground truth differentiable dynamic model. Empirically, the method by Karl et al. requires more samples to achieve reasonable performance and it has a larger variance. Moreover, the method’s ability to scale is unclear (it was neither demonstrated by Karl et al. 2017 nor we succeeded to scale it up to high dimensional image-based observations).
>
> 4. “Unclear They estimate empowerment for a state s_t, which now is used to train a policy. However, normally the reward is a value for s_t, a_t pairs, so essentially a function of s_t+1. Should empowerment also be computed for s_t+1?” -- Sorry that this was unclear before. Indeed reward for an (s_t, a_t, s_t+1) tuple is the empowerment value at s_t+1. This is consistent with line 10 of Algorithm 1, where s_0 is the start, and s_T is the end of a trajectory.
>
> 5. “Small notes Maximilian et al. should be Karl et al” -- thank you for catching this error. It is really embarrassing to misspell the names of our colleagues. We fixed it in our references.
>
> 6. “I am uncertain whether the Gaussian channel is a contribution to the research community. Is it only useful for research done on empowerment? Can it be used for other systems that use a VLB?” -- great point! One can map random variables to a latent space, where their dependencies are approximated to be Gaussian. Then, mutual information can be estimated by a variation of our method. This approach might be useful for representation learning and unsupervised learning by maximization of mutual information.
>
> Thank you again for your suggestions and review.

---

> > ### Comment · AnonReviewer1 · 2020-11-25
> > **Reproducibility**
> >
> > 1. In the updated paper I cannot find a link to a code repo.
> > 2. If you are estimating empowerment, then you are maximizing potential MI, not actual MI.

---

> > > ### Author Response · Authors · 2020-11-25
> > > **code repo & MI**
> > >
> > > 1. During the rebuttal we were asked by the fellow reviewers to perform additional experiments and comparisons between our method and the relevant prior art. (these additional experiments clarify the advantages of our methods, and position better our work). Some of them are finished just today.. Now we need to organize the code in order to make it public. We promise to do that asap, and anyone will be able to reproduce any of the experiments in the paper.
> > >
> > > 2. We estimate the maximal information between action sequences and future states, which is in accord with the formal  original definition of empowerment given by Eq. 1. Importantly, it was one of our goals to follow exactly the rigour formulation of empowerment which has very specific properties as we show in the Experiments section. Our "latent gaussain-channel empowerment" recovers the essential properties of empowerment in a broad range of dynamical systems. (We demonstrate that in high-dimensional mujoco environments in raw states, and in image-based observation as well). For our best knowledge, information theory does not provide a formal definition for 'potential mutual information '. The later is sometimes used as a descriptive/non-formal term in order to provide intuition and to emphasise the fact that we consider future states. In this sense, we agree that this information is potential mutual information about the future, which does not happen yet. We would be happy if you could suggest us whether we should rephrase/clarify anything in our descriptions in order to make the paper even more clear for the eventual readers. There is another term, 'predictive mutual information', [1] which is defined differently from empowerment, and deals with the future states as well. To avoid misunderstanding we clarify that we do not deal in the current work with 'predictive mutual information', but rather consider the classical empowerment (maximal mutual information between actions and future states).
> > >
> > > Thank you again for your review and valuable feedback!
> > >
> > > [1]  Bialek, William, Ilya Nemenman, and Naftali Tishby. "Predictability, complexity, and learning." Neural computation 13.11 (2001): 2409-2463.

---

### Official Review · AnonReviewer4 · 2020-11-02
**I lean towards rejection due to the clarity and form of the paper, however I welcome the authors to polish the paper during the rebuttal period.**

**Rating:** 7
**Confidence:** 3

**Review:**

The paper proposes an new algorithm to simultaneously estimate and maximise empowerment for achieving unsupervised stabilization. The method relies on the formulation of a dynamic system as a linear Gaussian channel. In this formulation, empowerment can be efficiently estimated by solving a line search problem. The authors propose to learn the channel matrix G(s) from samples and exploit then linear formulation to learn a corresponding policy maximising empowerment. These two steps are done in an alternating fashion until convergence. In the experiments the method is compared to previous approaches for unsupervised control via empowerment. The authors show convergence close to the true empowerment landscape and prove the sample efficiency and low variance of their method. Additionally, their method is capable of unsupervised control based on image observations.

Overall, I am torn between accepting and rejecting the paper and I lean towards rejection. I think it represents a good step  toward intrinsically motivated agents which can learn useful behaviour solely based on interaction with their environment, however, my major concern is about the clarity and the form of the paper. I think the description of the method is missing important details (see cons below) and I would like to see additional evaluations to support the claims of the paper. Overall I would recommend to polish especially the method and the experimental section before accepting it for publication.

Pros:
1. The method is well placed in the existing literature on empowerment and provides a thorough comparisons of past approaches.

2. The algorithm is novel and at the same time simple and very easy to understand. It relies on previous work for empowerment of linear Gaussian channels but frames the problem in a data-driven setting by learning the system dynamics from raw data.

3. The paper provides good experimental results which compare the method to existing approaches and show advantages when it comes to sample efficiency and convergence to the true empowerment landscape for the pendulum. Additionally, the method allows empowerment-based control from image observations for the first time.

Cons:
1. I miss important details in the paper:

    a) How is the parametrisation of G(s) with a neural network being done? Especially in the case of image based observations I would like more details.

    b) Connected to (a), please make it clear how the sequence of actions a_t^{H-1} is handled.

    c) You present an alternative to the SVD, that requires constraints. How do they enter the cost function or do you perform constraint optimization?

2. I think the claims of the paper are a bit bold. The method requires that the system dynamics can be formulated as a linear Gaussian channel, which requires a control affine system. Nonetheless the paper claims that the approach can handle arbitrary dynamics. Can you clarify how the approach would tackle non-affine systems (dx = f(x, u)). There is also no support for this claim in the experiments. Specifically, except for the ball-in-box, all experiments are on pendulum-like systems.

3. The subfigures 3g and 3h only show that the std of average empowerment is very low for the proposed method. I am missing an evaluation of the reward (empowerment) over time itself.

---

> ### Author Response · Authors · 2020-11-23
> **Revision posted with more precise claims and detailed explanations.**
>
> We thank the reviewer for acknowledging our contribution and providing suggestions for the presentation of our work. In this revision, we’ve concentrated on a thorough explanation of the method, and a more complete collection of experiments.
>
> 1. “How is the parameterization of G(s) with a neural network being done?” -- G(s) is the output layer of a 3-layer neural network, reshaped into a matrix. In the case of image observations, s is encoded to a latent space that supports reconstruction. In the revision, we’ve significantly improved the explanation of our architecture and training objectives in section 4.
>
> 2. “How the sequence of actions a_t^{H-1} is handled” -- For short-horizon (~10) classic-control environments, we simply concatenate the actions to form a long vector. For long-horizon or pixel-based tasks, the action sequences are encoded to latent actions by fully-connected neural networks.
>
> 3. “You present an alternative to the SVD, that requires constraints.” -- The alternative to SVD is not used in our implementation. We introduce it as a potential alternative if the SVD operation ever becomes a performance bottleneck, which we do not experience even for image-based empowerment estimation.
>
> 4. “I think the claims of the paper are a bit bold.” -- Overall, we have carefully checked our wordings for and toned down the claims to be more accurate. However, our method is indeed able to handle arbitrary dynamics, thanks to the encoding of observations and actions. In a newly added experiment, we show that our empowerment estimator successfully balances the Hopper at a standing position.
>
> 5. “I am missing an evaluation of the reward (empowerment) over time itself” -- Since the selected baselines for comparison all produce empowerment estimations on different scales, it is hard to compare their convergence directly. Thus, we choose to first compare the normalized empowerment at convergence as a check for correctness, then compare the relative standard deviation as a measure of convergence speed and stability.
>
> 6. We invite the reviewer to check out our improved experiment section as well. We added VIM (Mohamed & Rezende (2015)), an existing VLB empowerment estimator, as a baseline to better position our work among existing literature. In the revision, we also added unsupervised stabilization of the MuJoCo Hopper, which uses significantly more complicated underlying dynamics and a much longer horizon for empowerment estimation.
>
> We hope that the revision, along with the explanations puts our work in better shape for publication.

---

> > ### Comment · AnonReviewer4 · 2020-11-24
> > **Rebuttal response**
> >
> > I appreciate your response to the review.
> >
> > The revised paper includes the requested details and improves upon form and clarity of the paper, therefore I have increased my rating.
> >
> > The hopper experiment is a good addition and it boosts your claims. However, fundamentally hopper is a pendulum-like system. The observation process might be highly nonlinear, but the underlying dynamics are control affine and therefore it should be feasible to find a latent space in which the transition is linear. It would be interesting to see presented method on something comparable to a lorenz system.

---

### Decision · Program_Chairs · 2021-01-07
**Final Decision**

**Decision:**

Accept (Poster)

**Comment:**

The paper advocates empowerment for stabilising dynamical systems, the dynamics of which is estimated with Gaussian channels. Baseline comparisons have improved and that makes the experimental section good. While initial versions of the paper were problematic, all reviewer issues have been addressed and acceptance is almost unanimous.